# PeRFception: Perception using Radiance Fields

**Yoonwoo Jeong**[1][†]     **Seungjoo Shin**[1][†]     **Junha Lee**[1][†]     **Christopher Choy**[2]

**Animashree Anandkumar**[2,3]     **Minsu Cho**[1]     **Jaesik Park**[1]

**POSTECH**[1]     **NVIDIA**[2]     **Caltech**[3]

## Abstract

The recent progress in implicit 3D representation, i.e., Neural Radiance Fields (NeRFs), has made accurate and photorealistic 3D reconstruction possible in a differentiable manner. This new representation can effectively convey the information of hundreds of high-resolution images in one compact format and allows photo-realistic synthesis of novel views. In this work, using the variant of NeRF called Plenoxels, we create the first large-scale radiance fields datasets for perception tasks, called the **PeRFception dataset**, which consists of two parts that incorporate both object-centric and scene-centric scans for classification and segmentation. It shows a significant memory compression rate (96.4%) from the original dataset, while containing both 2D and 3D information in a unified form. We construct the classification and segmentation models that directly take this radiance fields format as input and also propose a novel augmentation technique to avoid over-fitting on backgrounds of images. The code and data are publicly available in `https://postech-cvlab.github.io/PeRFception/`.

## 1 Introduction

Over the last few years, advances in implicit representations have demonstrated great accuracy, versatility, and robustness in representing 3D scenes by mapping low dimensional coordinates to the local properties of the scene, such as occupancy [1, 2], signed distance fields [3, 4], or radiance fields [5, 6, 7]. They offer several benefits that explicit representations (e.g., voxels, meshes, and point clouds) could not represent: smoother geometry, less memory space for storage, novel view synthesis with high visual fidelity, to name a few. Thus, implicit representations have been used for 3D reconstruction [1, 2, 8, 9], novel view synthesis [5, 6, 7, 10, 11, 12, 13, 14, 15], pose estimation [16, 17, 18, 19], image generation [20, 21], and many more.

In particular, Neural Radiance Fields [5] (NeRF) and many follow-up works [10, 11, 12, 14, 15, 22] have shown that implicit networks can capture accurate geometry and render photorealistic images by representing a static scene as an implicit 5D function which outputs view-dependent radiance fields. They use differentiable volumetric rendering, a scene geometry, and the view-dependent radiance that can be encoded into an implicit network using only image supervisions. These components allow the networks to capture high fidelity photometric features, such as reflection and refraction in a differentiable manner unlike the conventional explicit 3D representations.

Given the success of the radiance fields, it is only natural to consider the radiance fields as one of the standard data representations for 3D and for perception. However, these novel representations, which can capture a scene with high fidelity, have not yet been used for perception tasks such as classification and segmentation. One of the main reasons is that there is no large-scale dataset that

---

† Authors contributed equally to this work.

36th Conference on Neural Information Processing Systems (NeurIPS 2022) Track on Datasets and Benchmarks.

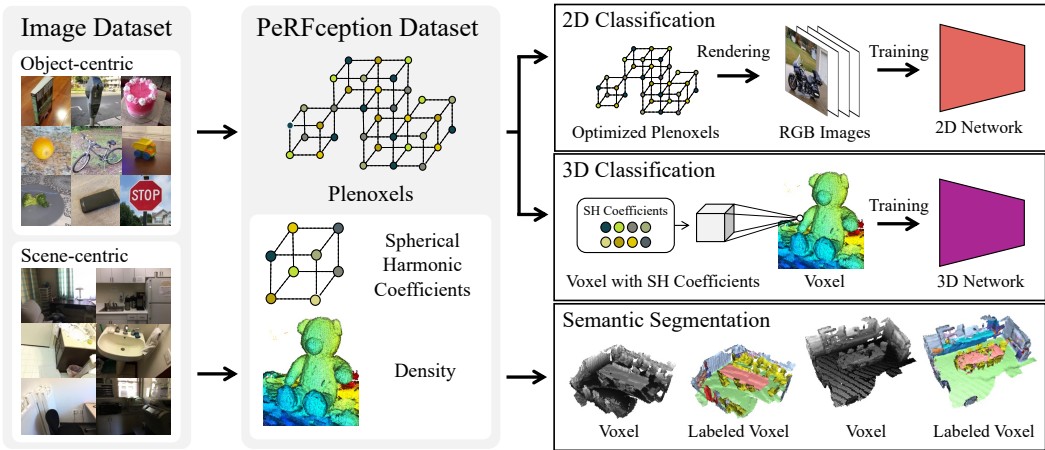

Figure 1: Overall illustration of PeRFception dataset with its applications. Our PeRFception dataset convey both visual (spherical harmonic coefficient) and geometric (density, sparse voxel grid) features in one compact format, it can be directly applied to various perception tasks, including 2D classification, 3D classification, and 3D segmentation.

allows training a perception system. Thus, in this work, we present the first large-scale radiance fields datasets to accelerate perception research.

Indeed, NeRFs have drawbacks that prevent the broad adoption of radiance fields as to the standard data format for 3D scenes and perception. First, training an implicit network is slow and can take up to days. Inference (volumetric rendering) also can take minutes, limiting the use of NeRFs in real-time applications. Second, the geometry and visual properties of a scene are implicitly encoded as weights in a neural network. These facts prevent an existing perception pipeline from processing the information directly. Third, implicit features or weights are scene-specific and are not transferrable between scenes. However, for perception, channels or features must have a consistent structure, such as RGB channels for images. For instance, if the order of channels is different from an image to an image, the image classification pipeline would not work properly.

Recent studies have resolved these limitations by adopting explicit sparse voxel grid geometry and basis functions for features. First, to tackle the slow speed, many works propose to use the explicit sparse voxel geometry, which reduces the number of samples along a ray by skipping empty space [10, 11, 12, 14, 15, 22]. Second, instead of using the implicit representations of weights of a network, directly optimizing features [14, 15, 22] assigned to explicit geometry reduces the time to extract features from a network. Lastly, for consistent features between scenes, which is crucial for perception or creating a scene with different objects in NeRF format, Yu et al. [14, 15] show that spherical harmonic coefficients can represent a scene as accurately as NeRFs while preserving consistent and structured features. In particular, Plenoxels [14] satisfy all criteria for data representation which supports fast learning and rendering while maintaining a consistent feature representation for perception and composition of scenes.

In this work, we adopt Plenoxels as the primary format for perception tasks and create both object-centric and scene-centric environments. We mainly use two image datasets and convert them into Plenoxels, the Common Object 3D dataset (CO3D) [23] and ScanNet [24], and name the converted datasets as PeRFception-CO3D and PeRFception-ScanNet, respectively. As the size of Plenoxels can be extremely large, we present a few techniques to compress the size of data and hyperparameters for each setup to maximize the accuracy while minimizing the data size.

We use the PeRFception datasets to train networks for 2D image classification, 3D object classification, and 3D semantic segmentation. We successfully train networks for each perception task, indicating that our datasets effectively convey 2D and 3D information together in a unified format. Moreover, we show that our representation allows more convenient background augmentation and sophisticated camera-level manipulation.

We summarize our contributions as follows:

- We introduce the first large-scale radiance fields datasets that can be readily used in downstream perception tasks, including 2D image classification, 3D object classification, and 3D scene semantic segmentation.
- We conduct the first comprehensive study of visual perception tasks that directly process the implicit representation. The extensive experiments show that our datasets effectively convey the information for 2D and 3D perception tasks.
- We provide the ready-to-use pipeline to generate the radiance fields datasets with fully automatic processes. We expect this automatic process allows generating a very large scale 3D dataset in future.

## 2 Related Work

### 2.1 Neural Implicit Representations

Representing a scene using an explicit representation such as voxels, meshes, or point clouds has been the most widely used format, but these are discrete and introduce discretization errors. Neural implicit representations, on the other hand, use a neural network to approximate the geometry or properties of a scene continuously [1, 2, 3, 25]. Mildenhall et al. [5] showed that neural radiance representation can generate high fidelity renderings with view-dependent illumination effects using multi-layer perceptrons. Many of recent studies extend such implicit representation to dynamic scenes [26, 27, 28, 29], conditional generation [30, 31, 32, 33], pose-free [16, 17, 19], and many more. In particular, research on efficient rendering [10, 12, 15, 22] has been one of the major directions since volumetric rendering could take minutes. Hedman et al. [10] propose to create sparse voxel grids after training to accelerate rendering. Similarly, Plenoctree [15] uses an octree data structure instead of sparse voxels for fast rendering. DVGO [22] and Plenoxels [14] also adopt the sparse voxel structure, and improve both inference and training time. INGP [12] proposes a multi-level hash encoding which enables the fast convergence. Recently, TensoRF [34] boosts both training and inference time by factorizing 3D radiance fields into lower dimensional vectors or matrices. In this work, we use Plenoxels for our data format since they have explicit geometry and consistent features in form of spherical harmonic coefficients.

### 2.2 3D Perception Datasets

Over the last decade, many public large-scale datasets of real objects for 3D perception have been published thanks to the advances in commodity sensors. In this section, we cover such large-scale 3D datasets for objects and scenes.

ShapeNet [35] and ModelNet [36] provide class and part annotations that are from synthetic CAD models. Early object-centric 3D datasets augment image datasets with 3D CAD model annotations. Pascal3D+ [37] and Objectron [38] contain 3D shapes that are matched with real-world 2D images containing objects; however, 3D models are chosen from approximately aligned 3D models, not precisely reconstructed from the corresponding 2D images. Redwood [39] is a large-scale object-centric RGB-D scan video dataset, where only a few categories include 3D models and camera poses. GSO [40] holds clear 3D models of real objects with textures, but missing physically rendered images. 3D-Future [41] provides synthetic CAD shapes with high-resolution informative textures developed by professional designers. CO3D [23] provides large-scale object-centric videos with camera poses and high-quality point cloud models. They assess quality of reconstructed 3D shapes using human-in-the-loop validation and marked 5,625 point clouds as successfully reconstructed. Recently, ABO [42] offers a dataset consisting of household object images and high-quality 3D models with 4K texture maps and full-view coverage. Professional artists manually designed its high-quality spatially-varying Bidirectional Reflectance Distribution Functions (BRDFs), indicating that the data generation processes were not fully automatic. We summarize the details of the aforementioned datasets in Table 1.

Many scene-centric 3D datasets use depth sensors to scan a section or an entire room and create dense annotations. SUN RGB-D [44] collected 13,355 RGB-D images, which are densely annotated with 2D polygons and 3D bounding boxes. However, it does not include camera parameters, which is essential information for surface reconstruction. NYUv2 [43] initially sparked interest for 3D scene understanding, with 464 indoor scans, 1,449 frames of which are annotated with 2D polygons for

Table 1: Specs for publicly available 3D datasets. "Real" denotes whether the objects are from real-world images , "Full 3D" for the availability of 3D geometries for all the objects, and "Multi-view" for multi-view images and corresponding real-world catalog images. ▲ is marked when the corresponding information is partially provided.

| Dataset | # Classes | # Objects | Real | Full 3D | Multi-view |
|---|---|---|---|---|---|
| ShapeNet[35] | 55 | 51K | ✗ | ✓ | ✓ |
| ModelNet[36] | 40 | 128K | ✗ | ✓ | ✗ |
| Pascal3D+[37] | 12 | 36K | ✓ | ✗ | ✗ |
| Redwood[39] | 9 | 2K | ✓ | ✗ | ✓ |
| Objectron[38] | 9 | 15K | ✓ | ▲ | ▲ |
| GSO[40] | ✗ | 2K | ✓ | ✓ | ✗ |
| 3D-Future[41] | 8 | 2K | ✗ | ✓ | ✗ |
| ABO[42] | 98 | 8K | ▲ | ✓ | ✓ |
| CO3D[23] | 51 | 19K | ✓ | ▲ | ✓ |
| PeRFception–CO3D | 51 | 19K | ✓ | ✓ | ✓ |

| Dataset | # Classes | # Scenes | Real | Full 3D | Multi-view |
|---|---|---|---|---|---|
| NYUv2 [43] | 894 | 464 | ✓ | ✗ | ▲ |
| SUN RGB-D [44] | 800 | ✗ | ✓ | ✗ | ▲ |
| SUN3D [45] | ✗ | 415 | ✓ | ✓ | ✓ |
| 2D-3D-S [46, 47] | 12 | 13 | ✓ | ✓ | ✓ |
| ScanNet [24] | 20 | 1,513 | ✓ | ✓ | ✓ |
| Matterport3D [48] | 40 | 90 | ✓ | ✓ | ✓ |
| Replica [49] | 88 | 18 | ✓ | ✓ | ✓ |
| PeRFception–ScanNet | 20 | 1,513 | ✓ | ✓ | ✓ |

semantic segmentation. SUN3D[45] is comprised of 415 RGB-D indoor video sequences in 254 different spaces; only eight sequences are annotated. Each sequence was captured densely, with a large number of frames collected. 2D-3D-S [46, 47] is an instance-level annotated large-scale indoor scene dataset. It offers diverse modalities of six indoor scenes in RGB images, depth maps, surface normals, 3D meshes, and point clouds. Currently, ScanNet [24] is the most popular large-scale indoor scene dataset that collected instance-level annotated 1,513 scans of RGB-D images and 3D data. Matterport3D [48] contains large-scale RGB-D images annotated with surface and semantic information. In particular, it covers a wide area of 90 building-scale scenes by capturing panoramic views, but it does not provide annotations. Replica [49] is a small but high-quality surface-annotated indoor scene reconstruction database. In this paper, we use two datasets, CO3D and ScanNet, to cover both object- and scene-centric dataset respectively.

We select CO3D for object-centric dataset since it consists of camera-annotated real-world images and has sufficient number of classes. In addition, we use ScanNet since it is one of the most popular 3D indoor dataset providing adequate number of data with rich annotations.

## 2.3 3D Perception Models

Unlike perceptions in the 2D domain, where the image is the de facto standard representation, there is no canonical representation for spatial 3D data. Existing explicit representations, such as voxels, meshes, and point clouds, target different aspects of data and have pros and cons. We categorize methods into two groups based on the input representation. Point-based methods [50, 51] directly consume the continuous 3D coordinates of point clouds or meshes using MLP and continuous/graph convolutions. Recent studies have tried to define custom convolution layers [52, 53, 54, 55, 56] upon the continuous coordinate space, or non-local operations [57, 58, 59]. Overall, these methods exhibit fast and simple processing, but it often requires a large computational cost due to neighbor search.

On the other hand, voxel-based methods discretize input coordinates into voxels, which introduces small quantization errors but allows fast neighbor lookup using a data structure. Specifically, recent advances in spatial sparse convolutions [60, 61, 62] that operates on sparse voxels utilize an efficient GPU hash table and require small memory footprint for neighbor search. It has shown successful adoption in many perception tasks, including semantic segmentation [61, 62, 63] , object detection [64], representation learning [65], and registration [66, 67, 68, 69]. We use the spatial sparse

convnets to create the first perception network on our PeRFception datasets due to its scalability in terms of memory footprint and computational cost.

## 3   Preliminary

Yu et al. [14] proposed a novel scene representation called Plenoxels that combines a sparse voxel grid for coarse geometry with spherical harmonic coefficients for radiance fields. Unlike conventional MLP-based NeRFs [5, 6, 10, 11], which use a single neural network to represent an entire scene, Plenoxels optimize coefficients of spherical harmonic in each non-empty voxel independently, which uses the same differentiable model for volumetric rendering described in NeRF [5] as follows:

$$\hat{C}(\mathbf{r}) = \sum_{i=1}^{N} T_i(1 - \exp(-\sigma_i\delta_i))\mathbf{c}_i, \quad \text{where } T_i = \exp(-\sum_{j=1}^{i-1} \sigma_j\delta_j), \tag{1}$$

where $T_i$ denotes light transmittance of a $i$-th sample, $\sigma$ is opacity, $\mathbf{c}$ is color, and $\delta$ is distance to the next sample on a ray $\mathbf{r}$. Plenoxels lookup the stored densities and spherical harmonic coefficients in the sparse voxel grids. For scenes with background, Plenoxels also use the lumisphere background representation to render the backgrounds.

We modified the official Plenoxels implementation to set the initial grid properly and hyperparameters. As the size of Plenoxels can be extremely large, we present a few techniques to compress the size of data and hyperparameters. More implementation details are in Section. 4.1, Section. 4.2, and the appendix.

Among many recent implicit radiance field representations [12, 14, 22, 34, 70, 71], we use Plenoxels [14] for our data representation for a few reasons—firstly, as we are creating radiance representations for O(10k) scenes, the training and rendering have to be fast for scalability. Ideally, the reconstruction process should take less than an hour, and our Plenoxel-based reconstruction takes 30 minutes per scene. Secondly, the representation should be able to capture unbounded scenes to represent the backgrounds. Lastly, we want features from the reconstruction to be consistent across scenes for 3D perception tasks. For instance, if we train a radiance field MLP per scene, each scene representation or feature would differ from others, and we cannot directly feed the features to a perception network without converting them to other consistent representations. Plenoxels uses explicit spherical harmonics features and density, which are consistent across scenes allowing us to train a perception network directly. We compare various radiance fields in Table 2 below. Note that Plenoxels [14] is the only representation that fits all the criteria.

Table 2: Properties of recent radiance fields representations.

| Method | Data structure | Density | Color | Training Time |
|---|---|---|---|---|
| PointNeRF [70] | Point Cloud | Explicit | Implicit | > 1 day |
| DVGO [22] | Dense Grid | Explicit | Hybrid | < 30 min |
| DVGO-v2 [71] | Dense Grid | Explicit | Hybrid | < 20 min |
| INGP [12] | Multi-level Hash | Hybrid | Hybrid | < 5 min |
| TensoRF [34] | Decomposed Grid | Explicit | Hybrid | < 30 min |
| Plenoxels [14] | Sparse Grid | Explicit | Explicit | < 30 min |

## 4   Generating PeRFception dataset

We generate two datasets PeRFception-CO3D and PeRFception-ScanNet to train perception networks.

### 4.1   PeRFception-CO3D

CO3D [23] is a large-scale object-centric dataset that contains multi-view observations of objects. It contains 18,669 annotated videos with a total 1.5 million of camera-annotated frames and 50 classes from MS-COCO [72], and images crowd-sourced from Amazon Mechanical Turk (AMT). It also provides reconstructed point clouds, generated by pretrained instance segmentation algorithm [73]

Table 3: Specs of CO3D and ScanNet, and our PeRFception-CO3D and PeRFception-ScanNet. **SH** denotes spherical harmonic coefficients, **D** for densities, and **C** for diffused color, "pcd" for point cloud. 3D-BG marks whether the 3D representation includes backgrounds of scenes. We note the number of frames in our proposed datasets as $\infty$ since our data representation is feasible to render frames from infinitely many camrea intrinsics and extrinsics.

| Dataset | # Scenes | # Frames | 3D Shape | Features | 3D-BG | Memory | Memory (Rel) |
|---|---|---|---|---|---|---|---|
| CO3D | 18.6K | 1.5M | pcd | **C** | ✗ | 1.44TB | $\pm 0.00\%$ |
| PeRFception-CO3D | 18.6K | $\infty$ | voxel | **SH,D** | ✓ | 1.33TB | $-6.94\%$ |
| ScanNet | 1.5K | 2.5M | pcd | **C** | ✗ | 966GB | $\pm 0.00\%$ |
| PeRFception-ScanNet | 1.5K | $\infty$ | voxel | **SH,D** | ✓ | 35GB | $-96.4\%$ |

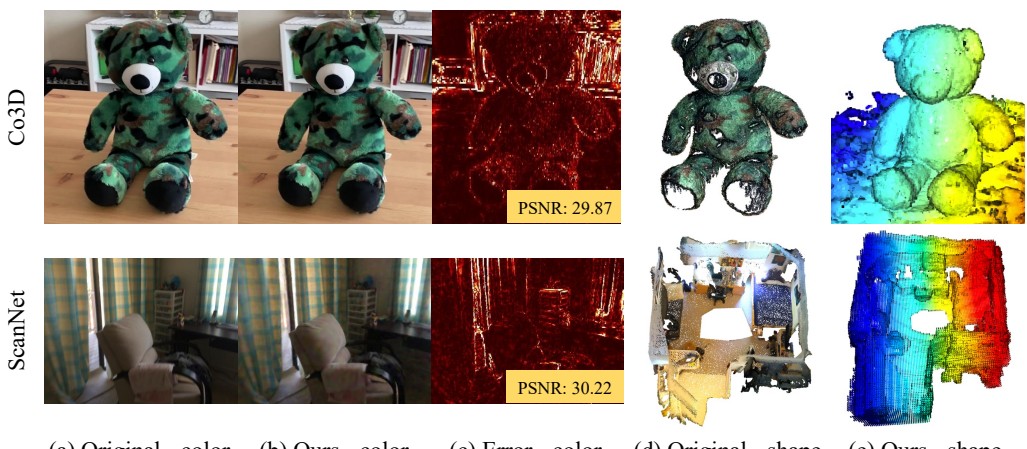

(a) Original - color    (b) Ours - color    (c) Error - color    (d) Original - shape    (e) Ours - shape

Figure 2: Visualization of a few example data of original datasets and our PeRFception datasets. From the source images and corresponding parameters, we successfully construct PeRFception datasets with both accurate geometry and photorealistic rendering. The color used in (e) is for visualization.

and COLMAP [74]. Although reconstructing depth and point clouds is automatic, its generation step still requires human-in-the-loop validation. When the amount of data increases, this human-in-the-loop validation becomes unsuitable. On the other hand, our 3D dataset generation does not require manual verification since image reconstruction qualities on unseen views are used as a proxy for reconstruction quality. We compare specs of the original CO3D and PeRFception-CO3D in Table 3.

**Data Generation.** We use the official implementation of Plenoxels [14] with a slight modification to the default configuration. We reduce the resolution of the background lumisphere from 1,024 to 512 and the number of background layers from 64 to 16. For sharper surface, we set the lambda sparsity value to $10^{-10}$, 10 times larger than the default configuration. A voxel grid is initialized with $128^3$ resolution and trained for 25,600 iterations. Then, it is upsampled once to $256^3$ resolution and trained for further 51,200 iterations. Before saving the data, we quantize the trained parameters to unsigned 8-bit integers to minimize for storage except for density values. For each scene, we first filter out defective images and uniformly sample 10% of the images as the test set to assess the rendering quality. The quantitative and qualitative results of the rendering quality are reported in Table 4 and Figure 2. More details are in the appendix.

## 4.2 PeRFception-ScanNet

ScanNet is a 3D-scanned indoor dataset that captures more than 1.5K indoor scenes with the commercial RGB-D sensors. It provides 3D reconstructed point clouds of scenes with semantic labels containing 20 common object classes, as well as the raw RGB-D frames with corresponding semantic masks and camera parameters. In our experiment, we follow the official data split and report the numbers on the validation split since the test set annotations are not publicly available. We compare specs of the original ScanNet and PeRFception-ScanNet in Table 3.

Table 4: Overall rendering qualities of PeRFception-CO3D and PeRFception-ScanNet on test set. For class-wise rendering scores are reported in the appendix.

| Dataset | PSNR(↑) | SSIM(↑) | LPIPS[1](↓) | Train Time | PSNR > 15 | PSNR > 20 | PSNR > 25 |
|---|---|---|---|---|---|---|---|
| PeRFception-CO3D | 28.82 | 0.8564 | 0.3451 | 21.6 min | 99.8% | 98.2% | 87.3% |
| PeRFception-ScanNet | 22.87 | 0.7912 | 0.4590 | 11.3 min | 98.3% | 68.1% | 34.0 % |

**Data Generation.** ScanNet videos are captured using handheld cameras where auto-exposure option is held. So, a fair number of frames contain motion blur which could lead to poor scene geometries. In practice, we generate the batches of rays before training and load them in CPU memory for efficient memory bandwidth utilization during training. Since the number of frames for each scene in ScanNet varies, we use uniformly-sampled 1,500 image frames at most. For the scenes with fewer than 1,500 images, we use them all after filtering out blurry images with a low variance of Laplacian [75]. Another characteristic of ScanNet is that, unlike object-centric datasets where cameras face inward, the images are captured from inside a room facing outward. These result in fewer images observing the same part of the space, which results in poor reconstruction of Plenoxel's geometry on ScanNet dataset. Specifically, the Plenoxel reconstruction artificially creates an excessive number of voxels in the empty space (i.e. floaters) to minimize the image reconstruction loss.

Instead, to supply an additional geometric prior to Plenoxel training, we initialized the voxel grid using the unprojected depth maps provided in ScanNet rather than starting from the dense voxel grid. However, since the provided depth maps of ScanNet are contaminated with noisy observations, we use TSDF integration to obtain smoother and complete scene surfaces and incorporate the connected component analysis to filter out the disconnected outlier points in the unprojected point clouds. This leads to stable and more accurate reconstruction and does not excessively generate floaters to minimize the rendering loss. The resulting PeRFception-ScanNet dataset occupies only 35GB in disk whereas the original video streams of ScanNet requires about 966GB disk space. This is a significant compression rate (96.4%), which emphasizes the accessibility of our representation as a dataset. Detailed dataset specs of the original ScanNet and PeRFception-ScanNet are reported in Table 3. We report the rendering quality on Table 4 and visualize the qualitative novel view renderings on Figure 2.

## 5 Experiment

We benchmark popular 2D image classification, 3D object classification, and 3D segmentation networks to demonstrate that our unified data format can be used for various perception tasks.

### 5.1 Classification on PeRFception-CO3D dataset

CO3D provides multi-view images of objects and 51 class labels for classification. We use the same class labels for classification of PeRFception-CO3D dataset. We adopt a few classification models for our dataset for both 2D [76] and 3D classification [62]. We split the dataset into the train, validation, and test set by scenes since the original CO3D does not provide such splits. We use $10\%$ of the scenes for validation set and $10\%$ for test set in each class. We use the same splits for 2D and 3D classification.

### 5.1.1 Implementation Details

All the 2D classification models are trained with the cross-entropy loss with the weight decay factor $10^{-4}$. Following the recommendations from [77], we utilize the label smoothing with $\epsilon = 0.005$, remove bias decay, and initialize weights of the batch normalization layers on the residual connections to 0. We use the SGD optimizer with momentum 0.9 and trained for 500 epochs with a batch size 64. For 50 epochs, we linearly warmed up the learning rate from 0 to 0.1 and decayed it using the cosine annealing scheduler. It takes up to a day for training using a single RTX 3090 GPU. We have benchmarked variants of ResNet (ResNet18, ResNet34, ResNet50, ResNet101, ResNet152),

---

[1]The LPIPS metric sometimes generates "nan" although their visual qualities are great enough. Only 0.01% of scenes are noted as failure.

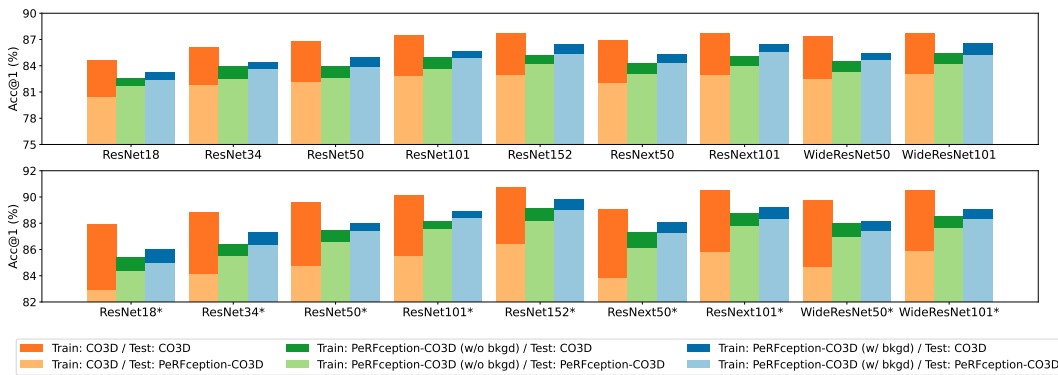

Figure 3: 2D classification accuracies (Acc@1) of the ResNet models trained either on CO3D or PeRFception–CO3D and evaluated either on CO3D or PeRFception–CO3D. * denotes the ImageNet [80] pretrained network. The models trained on PeRFception-CO3D dataset perform well on both CO3D test dataset and PeRFception-CO3D test dataset. Furthermore, the background augmentation of PeRFception-CO3D dataset improves the 2D classification performance. The score table is in the appendix.

ResNext [78] (ResNext50, ResNext101), and WideResNet [79] (WideResNet50, WideResNet101) networks.

For 3D classification, we train 3D version of ResNets [76] with varying depths that are implemented with spatially sparse convolutional layers [61, 62]. These networks directly take sparse voxels from Plenoxels as input. The Plenoxels consist of two components: coordinates of sparse voxels and their features (spherical harmonic coefficients and density values). To demonstrate the efficacy of such features in perception task, we train the networks by providing different input features. We use the SGD optimizer and set the initial learning rate as 0.1 for all experiments and decay it with the cosine annealing scheduler for 100K iterations with batch size 16 on a single RTX 3090 GPU. We augment the input data with both geometric augmentation (random rotation, coordinate dropout, random horizontal flip, coordinate uniform translation, and random scaling) and feature-level augmentation, random feature jittering. Further implementation details are in the appendix.

**Background Augmentation.** Plenoxels use both sparse voxels and lumispheres to render foregrounds and backgrounds respectively. In other words, we can render each of them separately or manipulate them to create various augmentations. Specifically, we create a novel augmentation that substitutes the background in a scene with backgrounds from other scenes while preserving the foreground object. We describe the composition of foregrounds and randomly selected backgrounds in the appendix. In addition, we visualize several background augmentation examples in the appendix.

### 5.1.2 2D Classification on PeRFception-CO3D

We train both the scratch and ImageNet [80] pretrained version of ResNet [76] variants on the original CO3D and PeRFception-CO3D to show that PeRFception-CO3D contains the same information as the original CO3D dataset. Using our random pose selection algorithm(described in the appendix), which discourages the selected pose to be extremely unobserved the train frames, we select 50 poses for each scene. As shown in Fig 3, the model trained with the original CO3D has a larger gap than the model trained with PeRFception-CO3D dataset. In addition, we observed using background augmentation is beneficial for improving generalizability of classification networks, especially performs the best when the augmentation is applied with probability $p = 0.5$. We conduct controlled experiments about the probability $p$ in the appendix. In addition, using a popular vision analysis tool GradCAM++ [81], we demonstrate that using background augmentation helps the model not to memorize the backgrounds in the appendix.

### 5.1.3 3D Classification on PeRFception-CO3D

PeRFception-CO3D provides a novel 3D data representation which we can directly feed into a network without explicit rendering. We train spatially sparse 3D networks on PeRFception-CO3D

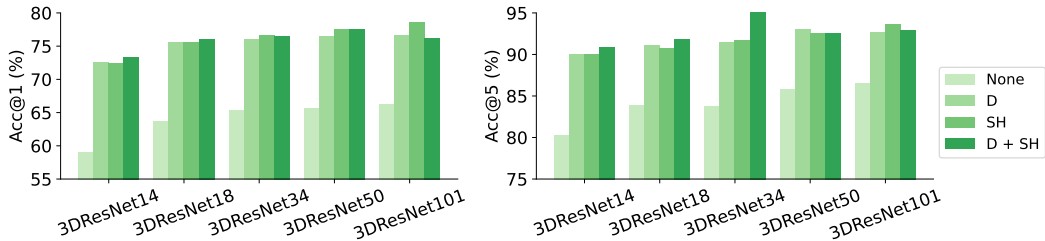

Figure 4: 3D classification performance of 3D ResNet [62] models on our PeRFception-CO3D. We visualize Acc@1 (Left) and Acc@5 (Right) score for each model and input features. "None" denotes the case where 3D classification with only geometric cue of sparse voxels. **D** denotes the density and **SH** denotes spherical harmonic coefficients.

and visualize the 3D classification accuracy on Figure 4. For each classification model, we utilize four types of input features that are in our Plenoxels representation: ones (None), densities (D), spherical harmonic coefficients (SH), and concatenation of spherical harmonic coefficients and densities (SH + D). One interesting observation is that using either density values or spherical harmonic coefficients as features improves performance much better. We conjecture this is because the density values provide information about where the model should focus more, and the spherical harmonic coefficients explicitly encode visual features.

## 5.2 Semantic Segmentation on PeRFception-ScanNet dataset

To further verify the fine-grained perception on the large-scale radiance fields data, we create and evaluate 3D semantic segmentation networks on our scene-centric PeRFception-ScanNet dataset. We assign semantic labels to each voxel by aligning the reconstructed PeRFception-ScanNet data with the provided ground truth point cloud data. Then, for each voxel of PeRFception-ScanNet data, we find the nearest point in the point cloud and when the distance is smaller than the predefined threshold (5cm for our experiments), we assign the class label of the nearest point to the voxel. Otherwise, we set the voxel label to IGNORE_CLASS.

Similar to 3D classification experiments, we use spatially sparse convolutional networks for prediction, but we use U-shaped convnets with varying depth and width for segmentation. Additionally, we employ the state-of-the-art large-scale 3D semantic segmentation model based-on transformer architecture, Fast Point Transformer (FPT) [63], to analyze the performance of the recent transformer-based 3D perception model on our dataset. For all networks, we trained for 60K iterations, with batch size 8, 2cm voxel size, SGD optimizer with initial learning rate 0.1, and cosine annealing scheduler. For FPT [63], we use voxel size of 5cm due to its extensive memory complexity. We train each network with different input features as same with the 3D classification in Sec 5.1 to analyze the effect of the plenoptic features to the 3D semantic segmentation task. We apply geometric augmentation (random rotation, random crop, random affine transform, coordinate dropout, random horizontal flip, random translation, elastic distortion), and feature-level augmentation (random feature jittering).

We use the standard experimental settings following [62], and report mean Intersection over Union (mIoU), mean per-point accuracy (mAcc) on the validation split in Figure 5, and scenewise statistics are in the appendix. We achieve up to 69.17% mIoU and 77.85% mAcc on the validation split of PeRFception-ScanNet, which shows that our PeRFception-ScanNet dataset has accurate geometry for networks to learn semantic of each object class. Consistent with 3D classification, the networks trained with spherical harmonic coefficients and density as input feature exhibit higher segmentation accuracy. These results indicate that the spherical harmonic coefficients and density values provide additional cues for fine-grained geometric perception as well. The qualitative visualization of semantic segmentation is shown in Figure 6. Additional quantitative and qualitative results are provided in the appendix.

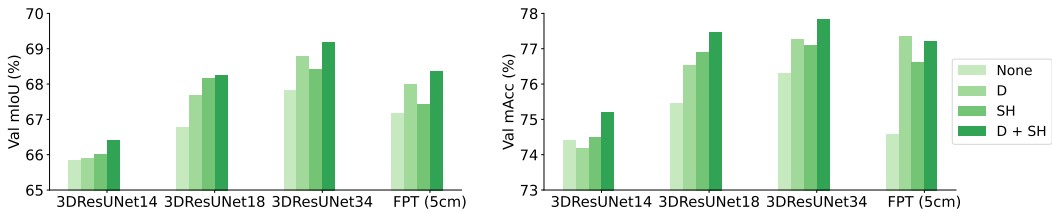

Figure 5: Evaluated semantic segmentation performance, mIoU (Left) and mAcc (Right), on PeRFception-ScanNet validation set with various input features. **D** denotes the density, **SH** denotes spherical harmonic coefficients.

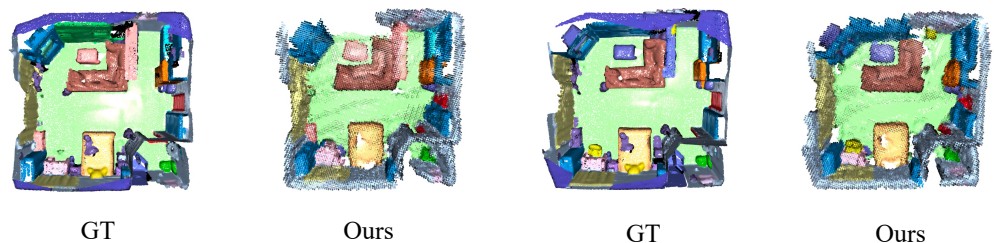

GT                    Ours                    GT                    Ours

Figure 6: Qualitative results of semantic segmentation on PeRFception-ScanNet dataset. (1st, 3rd columns) Ground truth point cloud with ground truth semantic labels, (2nd, 4th columns) Reconstructed sparse voxels with predicted semantic labels

## 6   Conclusion

In this work, we present the first perception networks for an implicit representation and conduct the comprehensive study of various visual perception tasks. To this end, we created two large-scale implicit datasets, namely PeRFception-CO3D and PeRFception-ScanNet, that cover object-centric and scene-centric environments, respectively. Extensive experiments with diverse perception scenarios, including 2D image classification, 3D object classification, and 3D scene semantic segmentation, show that our datasets effectively convey the same information for both 2D and 3D in a unified and compressed data format. This data format allows eliminating the need to separately store different data formats, 2D images and 3D shapes. Consequently, the required disk space for storage is reduced and the unified data format includes richer features. Furthermore, we propose a novel image augmentation method that was infeasible in image datasets. We expect our fully automatic pipeline should be a great candidate for establishing equally large datasets on 3D to tremendously large 2D image datasets, potentially enabling larger models to be trained.

**Limitation.** Plenoxels allow high-quality rendering for both indoor and outdoor scenes with fast training and rendering speed. However, the training step of Plenoxels strongly relies on calibrated camera information. The camera parameters would be inaccurately calibrated in the scenes when there are lots of symmetric or textureless patterns. Wrong camera information involves severe artifacts on rendered images, such as the occurrence of floater or geometrically deformed voxel shapes. We believe that jointly optimizing camera poses would be beneficial for improving the fidelity of our dataset. Our work opens up the potential of using radiance field representation in some conventional visual perception tasks and provides the first large-scale radiance field datasets that effectively convey both the 2D and 3D information. We expect future work relevant to more accurate and fast reconstruction could improve our work.

**Ethical Concerns.** Our work does not contain any serious ethical concerns of security threats or human rights violations. However, our dataset is capable of generating unseen views from multi-view images. Moreover, our object-centric dataset can be separated into foreground and background. It leads to background augmentation, which has a considerable effect on the 2D classification task. Thus, realistic fake videos whose camera trajectories are different from the original video or whose foreground and background are from different scenes can be generated from our dataset.

## Acknowledgments and Disclosure of Funding

This work was supported by the Institute of Information & Communications Technology Planning & Evaluation (IITP) grants (No.2021-0-02068: AI Innovation Hub (50%), No.2022-0-00290: Visual Intelligence for Space-Time Understanding and Generation based on Multi-layered Visual Common Sense (40%), No.2019-0-01906: AI Graduate School Program at POSTECH (10%)) funded by Korea government (MSIT).

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
