# OpenReview forum: "PeRFception: Perception using Radiance Fields"
_NeurIPS.cc/2022/Track/Datasets_and_Benchmarks — NeurIPS 2022 Datasets and Benchmarks _

### Official Review · Reviewer_f1Q2 · 2022-07-23
**Good paper that introduces the first large-scale implicit representation datasets for perception tasks**

**Rating:** 7
**Confidence:** 3
**Correctness:** 1. The dataset is constructed in a so…
**Clarity:** The paper is well written and easy to…

**Strengths:**

The paper introduces the first large-scale implicit datasets that can be readily used in downstream perception tasks. The dataset called PeRFception is both a timely and relevant contribution to the broader research community. Additionally, it shows a significant memory compression rate from the original dataset. Such a compression rate is beneficial for efficient research. The third strength of the paper is the pipeline to generate the implicit datasets automatically, which motivates to generate a very large scale 3D dataset in future.

**Weaknesses:**

1. Minimal discussion of the ethical or societal implications
2. Lack the comparisons of rendering qualities between PeRFception dataset and current popular 3D datasets in NeRF research.

**Additional Feedback:**

It would be better to provide the source code and full data link.

**Documentation:**

No code is provided. The documentation seems clear.

**Ethics:**

Does not appear to have any ethical concerns

**Relation To Prior Work:**

It is clearly discussed how this work differs from previous contributions.

**Summary And Contributions:**

The paper focuses on the datasets in implicit 3D representation. This work creates the first large-scale implicit representation datasets called PeRFception for perception tasks, including classification and semantic segmentation in NeRF. What's more, the paper provides a ready-to-use pipeline to generate the implicit datasets automatically.

---

> ### Author Response · Authors · 2022-08-22
> **Adding social and ethical implications, and comparisons with popular 3D datasets.**
>
> Thanks for your constructive comments.
>
> **1. Minimal discussion of the ethical or societal implications**
>
> We will add discussion of the ethical and social implications on our supplementary materials.
> Our dataset is capable of generating unseen views from multi-view images. Thus, realistic fake videos, whose camera trajectories are different from the original video, can be generated from our dataset.
>
> **2.  Lack the comparisons of rendering qualities between PeRFception dataset and current popular 3D datasets in NeRF research.**
>
> We will add quantitative rendering qualities on popular 3D datasets, namely, Tanks and Temples, in NeRF research for convenient comparison.

---

### Official Review · Reviewer_5TB9 · 2022-07-26
**The idea of using radiance field as a new data structure is interesting, but the contribution of data generation is not substantial.**

**Rating:** 6
**Confidence:** 4
**Clarity:** The paper is written clearly.

**Strengths:**

* This paper uses the radiance field, generated by a  NeRF-based method Plenoxels [14], as a novel data structure to replace traditional 2D or 3D data structure. It is an interesting attempt, since constructing a complete 3D structure of a scene is laborious. Directly generating implicit representation from input images for 3D perception will be really useful.
* Two datasets, PeRFception-Co3D and PeRFception-ScanNet, are introduced. The implicit representation gives another choice for 3D perceptron.
* The experiment shows that previous 2D/3D methods can directly apply on the novel data structure and the performance is well.

**Weaknesses:**

* The dataset generation is totally based on the method Plenoxels [14]. All data is directly stored from the intermediate data generated by the  Plenoxels, e.g. spherical harmonic coefficients, densities.  The contribution of this paper is not substantial enough.
* The density and color information are also generated by other NeRF-based methods [5, 6, 10, 11]. The 3D structure like points can be generated by Point-NeRF (https://arxiv.org/abs/2201.08845). The reason why we have to use the implicit representation of Plenoxels is not given. A thorough comparison between PeRFception and implicit data generated by methods mentioned before is necessary.
* The statement in L249, "PeRFception-Co3D contains the same information as the original Co3D dataset", can not be summarized from the results (Fig. 3). ResNet trained on PeRFception-Co3D has obvious performance drop compared with ResNet trained on the Co3D, when the test set is Co3D. Besides, ResNet trained on PeRFception-Co3D also has performance drop, when test set changes from Co3D to PeRFception-Co3D. I think there exists information loss after transforming original data to implicit one. A discussion about this is necessary.
*  A performance comparison between methods trained on ScanNet and  PeRFception-ScanNet is necessary, similar to Fig. 3.

**Additional Feedback:**

The main concern of mine is that the generation procedure of the dataset totally depends on Plenoxels. I will adjust my rating after the discussion with other reviewers and AC.

**Correctness:**

The claims in the submission are correct. The construction of the dataset and the instruction for using the dataset are clear.

**Documentation:**

The document of the dataset is detailed. The authors only provide a toy data download link where data is all from PeRFception-Co3D. I think the authors should provide a part of PeRFception-ScanNet dataset.

**Ethics:**

There is no ethic concern in my opinion.

**Relation To Prior Work:**

Relation to most of prior work has been discussed in the relation work. The relation to prior work Plenoxels needs further discussion.

**Summary And Contributions:**

This paper proposes the first large-scale neural radiance based dataset, which is generated by the Plenoxels [14]. Authors study visual perception tasks that directly process the implicit representation in their dataset. The pipeline for generating implicit is ready-to-use and provided for new data generation.

---

> ### Author Response · Authors · 2022-08-22
> **Response to the pointed issues**
>
> We appreciate for your constructive suggestions and comments.
>
> **1. The dataset generation is totally based on the method Plenoxels [14]. All data is directly stored from the intermediate data generated by the Plenoxels, e.g. spherical harmonic coefficients, densities. The contribution of this paper is not substantial enough.**
>
> Plenoxels has shown sufficiently great performance for rendering ScanNet and CO3D datasets. However, using the vanilla Plenoxel required 2\~3GB for each sequence, so we could not use them as our data format. Therefore, we have applied additional quantization and adjusted configurations while preserving the rendering quality. Accordingly, we have reduced the memory footprint of Plenoxel by more than 95% (from 2\~3GB to 50\~70MB) per scene.
>
> In addition, for ScanNet, we extend Plenoxels to be initialized from readily available geometry, reprojected depth images, or integrated TSDF volume. It allows more accurate photometric and geometry reconstruction. During the rebuttal period, we found that initializing Plenoxels with the geometry obtained with TSDF integration led to smoother geometry while maintaining the rendering quality. We’ll put the details and experimental results in the revised paper.
>
>
> **2. A performance comparison between methods trained on ScanNet and PeRFception-ScanNet is necessary, similar to Fig. 3.**
>
> Please note that, for the 3D semantic segmentation task, we can not directly compare the models trained on ScanNet and PeRFception-ScanNet since two datasets have different representations (e.g. Colored point clouds vs. 5D Radiance fields). Instead, we indirectly compare the segmentation accuracy by simply comparing the average mIoU scores in each domain, and the results show that we can achieve comparable segmentation accuracy on the PeRFception-ScanNet dataset while using the symmetric architectures. The results will be available soon in the revised paper.

---

### Official Review · Reviewer_nTsf · 2022-07-27

**Rating:** 6
**Confidence:** 4
**Correctness:** The dataset construction is technical…
**Clarity:** The paper reads fine, but the exposit…

**Strengths:**

1. Table 1 shows a clear comparison with existing datasets.

2. The proposed dataset can be used for 2D image classification, 3D object classification, and 3D scene semantic segmentation.

**Weaknesses:**

1. This paper creates an impression that the paper is good at data compression. I know this is not the intended purpose of this paper, but the paper reads like this.

2. The authors provided google drive links for URLs. People in some countries cannot access services provided by Google and hence cannot access this dataset.

**Additional Feedback:**

Please see the weakness section above.

**Documentation:**

URLs are provided with Google Drive links, which are not accessible for researchers in some countries.

**Relation To Prior Work:**

Related work is adequate.

**Summary And Contributions:**

This paper proposed a dataset called PeRFception for perception tasks such as classification and segmentation. The data is generated by a recent method called Plenoxels. This method allows a high memory compression rate while still maintaining 2D and 3D information.

---

> ### Author Response · Authors · 2022-08-22
> **Improving exposition and allowing better accessibility**
>
> Thank you for your valuable suggestions.
>
> **1. This paper creates an impression that the paper is good at data compression. I know this is not the intended purpose of this paper, but the paper reads like this.**
>
> We attach our response regarding this issue in the common response.
>
> **2. The authors provided google drive links for URLs. People in some countries cannot access services provided by Google and hence cannot access this dataset.**
>
> Thanks for your great suggestion concerning accessibility. We have moved our data to OneDrive, which is accessible in any country. Moreover, we provide convenient utilization for downloading OneDrive files. Please refer to the common response.

---

> > ### Comment · Reviewer_nTsf · 2022-08-25
> > **Reviewer response**
> >
> > Thank you for addressing my comments. I decided to increase my score.

---

### Official Review · Reviewer_FBBw · 2022-07-27
**A new dataset representation using neural radiance field, but the potential for practical usage is limited.**

**Rating:** 6
**Confidence:** 3
**Correctness:** Yes.

**Strengths:**

* The first work to use implicit representation for a large-scale dataset.
* The compression rate for the ScanNet dataset is impressive.
* In the proposed dataset, more data augmentation can be performed (e.g., render novel views, substitute the background).

**Weaknesses:**

I like the paper's motivation, and I believe there is some potential for an implicit representation of the dataset.
However, I have some concerns about the effectiveness of the proposed method for practical usage.

* Though the data representation is unified, it seems it can not help to simplify the training process or boost performance. Especially for 2D tasks, the images should be rendered first (or generated during the training process). The rendering may also slow down the training process, which the authors do not report. There is no such step if using the original dataset. Besides the rendering time, the reconstruction error is also introduced. For the 3D task, this problem is alleviated by learning from the implicit representation directly, but it will present another problem. The network architecture of the existing method should be re-designed to adapt to this data format.

* Data compression is the core advantage of the implicit representation. But the result on the "object-centric" dataset is not good enough. As the image quality will decrease, I wonder if it is worth this trade-off.

* With background augmentation, the accuracy result (test on the original test set)  is still lower than trained with the original dataset, as shown in Fig.3.

* With the drawback of implicit representation, it seems hard to represent synthetic datasets like 3D-FUTURE/3D-FRONT well. Specifically, the scene editing (e.g., material, lighting) and domain randomization technique are not supported. Also, the advantage of generating accurate ground truth (e.g., noise-less depth) is hard to preserve.

Some minor concerns:
* ResNet is used for all experiments. Experimenting with some SOTA methods in the classification and semantic segmentation task is better.

* Fig.3 needs more discussion. It is desirable that the result trained on PeRFception has a lower accuracy than the original dataset, showing a domain gap caused by reconstruction error. However, when evaluating using the test set of PeRFception, the result (i.e., shallow green and blue) that trained on PeRFception (expected to have a lower domain gap with the test set) is still lower than the result on the original test set (i.e., dark green and blue). Please discuss.

* The choice of Plenoxels needs more discussion. The author says it has explicit geometry and consistent features, but other methods like INGP can also obtain geometry and be trained faster.

**Additional Feedback:**


* The authors state in L311 that there are discussions on the limitation in the supplementary document. However, I did not find it.

* Table.2: the abbreviation "pcl" for point-cloud should be noted in the caption.

* L194: It seems a bit confusing, and more details are needed. How many frames? Why need to saturate the CPU memory?

**Clarity:**

Overall, the paper is clear and well-structured.


**Documentation:**

Both the maintenance statement and the license are provided. But the documentation for the dataset seems a bit insufficient. The documentation with a more formal format like "datasheet" is preferred.

The authors also provided the toy dataset for evaluation and the corresponding code to view the 3D shape. The link to the full dataset is required, though it is not ready yet.

Though the author stated as a contribution that "ready-to-use pipeline to generate the implicit datasets with fully automatic processes", the code is also not provided yet.

Since the author also states the dataset and the code will be released, I think it's not a major problem.

**Ethics:**

The dataset this works based upon contains the common objects (CO3D) and interior scenes (ScanNet). I think there are no ethical problems.

**Relation To Prior Work:**

Some minor problems:

* "Co3D" -> "CO3D"
* Table.1 only shows the specs of object-centric datasets, a similar comparison for the scene-centric dataset is preferred.
Also, the 3D-Future dataset is listed in the table, but not discussed in the main text.
* L117: "Both datasets do not include camera parameters". The NYUv2 dataset seems to contain camera parameters, as said on its [website](https://cs.nyu.edu/~silberman/datasets/nyu_depth_v2.html), "camera_params.m - Contains the camera parameters for the Kinect used to capture the data.".

**Summary And Contributions:**

This work introduces the recent implicit 3D scene representation method for constructing datasets. This new representation can unify 2D images and 3D scene datasets into one compact format. Specifically, the authors evaluate this idea by building two datasets upon the CO3D and ScanNet datasets. Then the authors tested the accuracy with multiple ResNet variants to show lower memory and the availability of the proposed datasets.

---

> ### Author Response · Authors · 2022-08-22
> **Enhancing the manuscript, response to the pointed issues (1/2)**
>
> Thanks for your constructive suggestions.
>
> **1-1. “Though the data representation is unified, it seems it can not help to simplify the training process or boost performance. Especially for 2D tasks, the images should be rendered first (or generated during the training process). The rendering may also slow down the training process, which the authors do not report. There is no such step if using the original dataset. Besides the rendering time, the reconstruction error is also introduced.”**
>
> Rendering images from PeRFception-CO3D during the training time -- which could provide a wide range of augmentations -- is computationally expensive. For example, rendering a scene with 224 by 224 resolution takes 70.4ms, and if we use 4096 batch size, data generation alone takes 288s. Therefore, we pre-render images before training. However, PeRFception-CO3D allows us to render from arbitrary viewpoints, intrinsics change, various background augmentations, and other augmentations such as adding other occluders. This data format allows completely new ways to augment datasets that we have not covered and would open new training schemes. In addition, as shown in our ViT experiments, our augmentation actually boosts the CO3D 2D classification performance.
>
>
> **1-2. “For the 3D task, this problem is alleviated by learning from the implicit representation directly, but it will present another problem. The network architecture of the existing method should be re-designed to adapt to this data format.”**
>
> Since our representation can model the geometry with sparse voxel grids, any architectures that are capable of processing point clouds or voxels can be applied to our datasets. Although some networks that are capable of processing 3D coordinates (e.g. PointNet) can not utilize the plenoptic features (SH, density), 3D convolutional networks (e.g. submanifold sparse convolution, kernel-point convolution, etc), and the recent Transformer-based models (e.g. Point Transformer, Fast Point Transformer) can be easily adapted to our representation without any modification.
>
>
> **3. It seems hard to represent synthetic datasets like 3D-FUTURE/3D-FRONT well. Specifically, the scene editing (e.g., material, lighting) and domain randomization technique are not supported. Also, the advantage of generating accurate ground truth (e.g., noise-less depth) is hard to preserve.**
>
> Although synthetic datasets are beneficial for acquiring accurate ground truth, it involves a domain gap with real-world 3D datasets. Recent NeRF papers [1,2,3,4] show that it is possible to change the lighting by learning the material properties and re-render. Also, scene editing on real 3D datasets can be future work on our dataset.
>
>
> **4. Typo: Co3D -> CO3D**
>
> Thanks. We will update it.
>
>
> **5. Table.1 only shows the specs of object-centric datasets, a similar comparison for the scene-centric dataset is preferred. Also, the 3D-Future dataset is listed in the table, but not discussed in the main text.**
>
> Thanks for the comment. we will add the table for the scene-centric datasets.
>
>
> **6. L117: "Both datasets do not include camera parameters". The NYUv2 dataset seems to contain camera parameters, as said on its website, "camera_params.m - Contains the camera parameters for the Kinect used to capture the data.".**
>
> Thanks for the comment. We will fix it.
>
> **7. The authors state in L311 that there are discussions on the limitation in the supplementary document. However, I did not find it.**
>
> Thanks for pointing this out. We will add the following limitations of our work in the final version.
>
> *Limitation*
>
> Plenoxels allow high-quality rendering for both indoor and outdoor scenes with fast training and rendering speed. However, the training step of Plenoxels strongly relies on calibrated camera information. The camera parameters would be inaccurately calibrated in the scenes when there are lots of symmetric or textureless patterns. Wrong camera information involves severe artifacts on rendered images, such as the occurrence of floater or geometrically deformed voxel shapes. We believe that jointly optimizing camera poses would be beneficial for improving the fidelity of our dataset. Our work opens up the potential of using radiance field representation in some conventional visual perception tasks and provides the first large-scale radiance field datasets that effectively convey both the 2D and 3D information. We expect future work relevant to more accurate and fast reconstruction could improve our work.

---

> > ### Author Response · Authors · 2022-08-22
> > **Enhancing the manuscript to clarify the details (2/2)**
> >
> > **8. Table.2: the abbreviation "pcl" for point-cloud should be noted in the caption.**
> >
> > Thanks for the comment. We will update the description about the abbreviation “pcl.”
> >
> >
> > **9. L194: It seems a bit confusing, and more details are needed. How many frames? Why need to saturate the CPU memory?**
> >
> > In practice, we generate the batches of rays before training and load them in CPU memory for efficient memory bandwidth utilization during training. Since the number of frames for each scene in ScanNet varies, we use uniformly-sampled 1,500 image frames at most. For the scenes with fewer than 1,500 images, we use them all after filtering out blurry images. We found that this is a reasonable setup for the hardware environment that we used.
> >
> >  **Reference**
> >
> > [1] Verbin, Dor, et al. "Ref-nerf: Structured view-dependent appearance for neural radiance fields." arXiv preprint arXiv:2112.03907 (2021).
> >
> > [2] Bi, Sai, et al. "Neural reflectance fields for appearance acquisition." arXiv preprint arXiv:2008.03824 (2020).
> >
> > [3] Srinivasan, Pratul P., et al. "Nerv: Neural reflectance and visibility fields for relighting and view synthesis." Proceedings of the IEEE/CVF Conference on Computer Vision and Pattern Recognition. 2021.
> >
> > [4] Zhang, Xiuming, et al. "Nerfactor: Neural factorization of shape and reflectance under an unknown illumination." ACM Transactions on Graphics (TOG) 40.6 (2021): 1-18.

---

> > > ### Comment · Reviewer_FBBw · 2022-08-23
> > > **Response to authors**
> > >
> > > Thanks for the authors' response.
> > > I think some concerns (e.g., results on the "object-centric" dataset and background augmentation, the choice of Plenoxels) are addressed, I decide to raise my rating.

---

### Official Review · Reviewer_scvN · 2022-07-28
**comprehensive study on perception tasks on one type of pretrained explicit radiance fields**

**Rating:** 7
**Confidence:** 4

**Strengths:**

- large-scale training of plenoxels on co3d and scannet, effectively compressing the video sequences into *.plenoxels format
- comprehensive study of perception tasks on those pretrained plenoxels, shows reasonable good results
- background augmentation and camera manipulation can be enabled with neural rendering, which are non-trivial for traditional 2d/3d data format

**Weaknesses:**

- for nerf-like models to be real candidate to compress data (data format), we probably need more study on the rate-distortion-compute trade-off, which is lacking in this work. e.g. the compression scheme (quantization) is quite vanilla.
- the paper claims implicit representation dataset, but plenoxels are more like explicit representation, please make it more clear in abstract and paper
- we can think of perception on trained plenoxels as two-step processes: first compress images into plenoxels, then use the compressed features for perception tasks. One can argue the compression step may lose important information for downstream perception tasks. Thus end-to-end perception models should still achieve better accuracy.


**Additional Feedback:**

maybe provide some comparison to the stoa results on those perception tasks benchmarked in this paper

**Clarity:**

- can you provide more details on the visuals in fig 2 (e)? what is the color for?
- Table 2 caption: C for diffused color? what do you mean?
- line 194 - saturate GPU memory?
- maybe put training time in the main time, 30min per scene

**Correctness:**

- fig 3: there is 4-5% accuracy@1 gap for models trained on co3d and test on co3d / perfception-co3d, which indicates that plenoxels lose info for classification tasks? Also the gap is larger for models trained on PeRFception-Co3D. It makes me doubt the claim "PeRFception-Co3D contains the same information as the original CO3D dataset"
- fig4: why use both density and SH makes 3d classification accuracy drop for 3dResNet101?
- table 2 & 3: scannet compression ratio is much larger than co3d, with much worse psnr (23 vs 28). This is expected from rate-distortion trade-off. The paper should make it clear. Can you adjust the trade-off for scannet? e.g. better psnr with larger model sizes.

**Documentation:**

there are detailed instructions about datasets, future maintaining plan, and license.

**Relation To Prior Work:**

maybe add a few papers related to compression with implicit neural repr. e.g.
- Dupont, Emilien, et al. "Coin: Compression with implicit neural representations." arXiv preprint arXiv:2103.03123 (2021).

quite relevant work (maybe concurrent)
- Wang, Clinton J., and Polina Golland. "Deep Learning on Implicit Neural Datasets." arXiv preprint arXiv:2206.01178 (2022).

**Summary And Contributions:**

This work explores the potential of using explicit radiance fields as a unified compressed format for 2d image sequences and 3d scenes. It presents two datasets of pretrained plenoxels of CO3D (objects) and ScanNet (indoor scenes), which are used for benchmark perception tasks such as 2d/3d classification and 3d semantic segmentation, through either rendering 2d images or directly using its feature grids. It hopes to inspire future work to build larger scale 3d datasets and train bigger vision models.

---

> ### Author Response · Authors · 2022-08-22
> **More studies about quantization methods for an efficient data representation (1 / 3)**
>
> Thank you for your suggestions. We provide additional experiments on resolution-memory trade-offs, quantization methods, and progressive scaling methods. We will update the manuscript to reflect the changes.
>
> **1. “We probably need more study on the rate-distortion-compute trade-off, which is lacking in this work. e.g. the compression scheme (quantization) is quite vanilla.”**
>
> We additionally explore quantization methods and resolution-memory trade-offs to search for the optimal configuration. All the experiments will be updated in the paper.
>
> **[PeRFception-CO3D]**
>
> **Experiment 1) Resolution-memory trade-off**
>
> We compare the reconstruction qualities by varying the resolution: 64, 128, 256, and 384. We follow the setup from our paper and measure the average memory footprint and error metrics (PSNR, SSIM, LPIPS) per scene on the randomly selected subset of our PeRFception-CO3D dataset.
>
>  | Resolution | Memory (MB) |  PSNR  |   SSIM  |  LPIPS  |
> |:----------:|:-----------:|:------:|:-------:|:-------:|
> |     64     |     38.0    |  26.56 |  0.7464 | 0.4827  |
> |     128    |     47.2    |  29.39 |  0.8081 |  0.4017 |
> |     256    |     63.2    | 30.81  | 0.8551  | 0.3353  |
> |     384    |    161.2    |  31.03 |  0.8619 | 0.3202  |
>
>
> **Experiment 2) Quantization Methods**
>
> We further explore several quantization methods and compression bits that can efficiently quantize our data representations. “ours(SH)” applies our quantization on spherical harmonics coefficients only; “ours(SH+D)” applies our quantization to both spherical harmonics coefficients and densities. “Clipping” clips the feature values to a heuristically searched interval; for instance, density should be non-negative values. We will elaborate on thorough details about the compression methods in the supplementary materials. We additionally seek the optimal bit for each compression method. For convenience, we denote each method and bit as [Method]-[bit].
>
> | Quantization Method | Bit | Memory (MB) |  PSNR |  SSIM  |  LPIPS  |
> |:-------------------:|:---:|:-----------:|:-----:|:------:|:-------:|
> |      ours(SH)      | 16 | 114.5 | 30.65 | 0.8548 | 0.3363 |
> |       ours(SH)      |  8  |     63.2    | 30.81 | 0.8551 |  0.3353 |
> |       ours(SH)      |  4  |     37.4    | 27.48 | 0.7932 | 0.4512  |
> |       ours(SH)      |  2  |     24.8    | 14.69 | 0.5325 |  0.6797 |
> |     ours(SH+D)      |  8  |     60.8    | 30.53 | 0.8546 | 0.3369  |
> |      ours(SH+D)     |  4  |     34.8    | 23.07 | 0.7238 |  0.5166 |
> |     ours(SH + D)    |  2  |     21.8    | 14.65 | 0.5280 |  0.9981 |
> |       Clipping      |  8  |     62.6    | 17.78 | 0.7227 |  0.4879 |
> |       Clipping      |  4  |     37.4    | 17.72 | 0.6575 |  0.5502 |
> |       Clipping      |  2  |     24.4    | 16.66 | 0.6476 |  0.5881 |
>
>
> We observed that our original quantization method with 8 bit compression, i.e., ours(SH)-8bit, shows the best performance and has reasonable memory requirement.
>
> **Experiment 3) Progressive Scaling**
>
> Finally, we explore the effect of *progressive scaling*, i.e. upsampling and pruning step of Plenoxel to progressively increase the resolution on memory footprint and rendering quality. We consider two progressive scaling methods: *weight-based scaling* and *density-based scaling*. According to Plenoxels[2], *weight-based scaling* applies threshold to the maximum weight $T_i(1-exp(1-\sigma_i\delta_i))$ of each voxel over all training rays, whereas *density-based scaling* directly prunes voxels by their density values. For more details, please refer to the supplementary materials of Plenoxels[2]. For each method, we change the threshold of each scaling method to find a good adjustment between memory usage and rendering quality.
>
> | Progressive Scaling         | Threshold | Memory(MB) |  PSNR |  SSIM  |  LPIPS |
> |-----------------------------|-----------|:----------:|:-----:|:------:|:------:|
> |            density            |     5     |    69.0    | 30.66 | 0.8554 | 0.3352 |
> |            density            |     10    |    66.4    | 30.79 | 0.8550 | 0.3354 |
> |         density (ours)        |     20    |    63.2    | 30.81 | 0.8551 | 0.3353 |
> |            density            |    100    |    49.8    | 30.47 | 0.8547 | 0.3537 |
> | weight (Plenoxel - default) |    1.28   |    75.8    | 30.81 | 0.8557 | 0.3333 |
> |        weight - 2.56        |    2.56   |    73.0    | 30.71 | 0.8560 | 0.3335 |
>
>
> We select the “density-based scaling” method as our progressive scaling method with a sigma threshold 20, which has shown the best trade-off between memory footprint and rendering quality.

---

> > ### Author Response · Authors · 2022-08-22
> > **Response to the pointed issues (2/3)**
> >
> >
> > **[PeRFception-ScanNet]**
> >
> > We also analyze the *resolution vs. rendering quality* trade-off on our PeRFception-ScanNet dataset. To measure the trade-off rates, we train Plenoxels with lower (128) and higher (512) resolutions than the default configuration (256) on a randomly selected subset of ScanNet scenes. The results are reported in the below table. PCTL stands for percentile.
> >
> > | Reso | Mem (GB) | Avg. PSNR  | 50th PCTL | 75th PCTL | 90th PCTL | 95th PCTL |
> > |---|---|---|---|---|---|---|
> > | 128 | 28.7 | 23.20+-3.69 | 23.59 | 26.14 | 27.94 | 29.12 |
> > | 256 | 43.8 | 23.01+-3.96 | 23.38 | 26.10 | 28.02 | 29.20 |
> > | 512 | 113.8 | 22.94+-4.26 | 23.22 | 26.35 | 28.34 | 29.49 |
> >
> > As reported in the table, there is no direct correlation between the resolution and the average rendering quality on ScanNet reconstruction. This could be attributed to various non-trivial factors. We visualized the histogram of PSNR distribution on the ScanNet dataset in Figure a.3 in the appendix . The X-axis represents the PSNR score, and Y-axis represents the percentage of scenes. Note that the PSNR distribution of the higher resolution Plenoxel reconstructions exhibits fat-tailed distribution, whereas the lower resolution reconstructions show the long-tailed distribution. We speculate that this is due to the fact that higher resolution reconstruction results in each voxel learning spherical harmonics parameters from fewer rays. Thus, errors in camera parameters or motion blur would result in larger errors for smaller voxels as parameters are learned from fewer rays. Thus, the rendering quality increases for the scenes with accurate camera poses and less motion blur, while it decreases for noisy scenes. We conjecture that higher resolution reconstruction would yield better performance if we have high-resolution images with accurate camera poses.
> >
> >
> >
> > **2. “The paper claims implicit representation dataset, but plenoxels are more like explicit representation, please make it more clear in abstract and paper”**
> >
> > That’s a good point. We will clearly state that our datasets have *explicit* representation.
> >
> >
> > **3. “Fig4: why use both density and SH makes 3d classification accuracy drop for 3dResNet101?”**
> >
> > As reported in the supplementary materials, the performance of the 3DResNet101 model with SH falls within the 68%(\pm \sigma) confidence interval of the performance of the 3DResNet101 model with SH + D. We speculate that the number of training iterations for the largest 3DResNet101 models is not sufficient for them to reach their maximum accuracy.
> >
> >
> >
> > **4. “Table 2 & 3: scannet compression ratio is much larger than co3d, with much worse psnr (23 vs 28). This is expected from rate-distortion trade-off. The paper should make it clear. Can you adjust the trade-off for scannet? e.g. better psnr with larger model sizes.”**
> >
> > There are a few factors that enable the significant compression in PeRFception-ScanNet. First, the RGB-D video frames of the ScanNet dataset are much more redundant than the sparse input views of CO3D-v1. Second, there is no need to represent the backgrounds for ScanNet since all the walls, floors, and ceilings should be considered as the foreground geometry. Lastly, we initialize the Plenoxel voxel grids with the 3D geometry, and it prevents the output sparse voxel grid after training from having an excessive number of voxels in the void and the interior space of the objects.
> >
> > As discussed earlier, the tendency between resolution and the rendering quality varies depending on the degree of the motion blur and camera noises. We believe that the joint optimization of camera poses to compensate for noisy initial poses[6,7] would be beneficial for better representation quality for our dataset.
> >
> >
> > **5. “Can you provide more details on the visuals in fig 2 (e)? what is the color for?”**
> >
> > In Fig.2(e), we visualize the density value of our dataset to demonstrate that it contains accurate geometry. The color of Fig.2(e) is only for visual convenience. Thus, there is no special significance.
> >
> >
> > **6. “Table 2 caption: C for diffused color? what do you mean?”**
> >
> > Thank you for pointing out that this expression is confusing. We intended to indicate that C is the RGB color value captured by the camera. We will change it correctly.
> >
> >
> > **7. L194 - saturate GPU memory?**
> >
> > In practice, we generate the batches of rays before training and load them in CPU memory for efficient memory bandwidth utilization during training. Since the number of frames for each scene in ScanNet varies, we use uniformly-sampled 1,500 image frames at most. For the scenes with fewer than 1,500 images, we use them all after filtering out blurry images. We found that this is a reasonable setup for the hardware environment that we used.

---

> > > ### Author Response · Authors · 2022-08-22
> > > **Response to the pointed issues**
> > >
> > > **8. Maybe put training time in the main table, 30min per scene**
> > >
> > > Thanks. We will add more details about this part.
> > >
> > > **9. More related work**
> > > We will update the suggested work on the Related Work session.
> > >
> > > - [1] Dupont, Emilien, et al. “Coin: Compression with implicit neural representations.” arXiv preprint arXiv:2103.03123 (2021).
> > > - [2] Wang, Clinton J., and Polina Golland. “Deep Learning on Implicit Neural Datasets.” arXiv preprint arXiv:2206.01178 (2022).
> > >
> > >  **Reference**
> > >
> > > [1] Dupont, Emilien, et al. "Coin: Compression with implicit neural representations." arXiv preprint arXiv:2103.03123 (2021).
> > >
> > > [2] Fridovich-Keil, Sara, et al. "Plenoxels: Radiance Fields Without Neural Networks." Proceedings of the IEEE/CVF Conference on Computer Vision and Pattern Recognition. 2022.
> > >
> > > [3] Dosovitskiy, Alexey, et al. "An image is worth 16x16 words: Transformers for image recognition at scale." arXiv preprint arXiv:2010.11929 (2020).
> > >
> > > [4] Tolstikhin, Ilya O., et al. "Mlp-mixer: An all-mlp architecture for vision." Advances in Neural Information Processing Systems 34 (2021): 24261-24272.
> > >
> > > [5] Park, Chunghyun, et al. "Fast Point Transformer." Proceedings of the IEEE/CVF Conference on Computer Vision and Pattern Recognition. 2022.
> > >
> > > [6] Azinović, Dejan, et al. "Neural RGB-D surface reconstruction." Proceedings of the IEEE/CVF Conference on Computer Vision and Pattern Recognition. 2022.
> > >
> > > [7] Jeong, Yoonwoo, et al. "Self-calibrating neural radiance fields." Proceedings of the IEEE/CVF International Conference on Computer Vision. 2021.

---

### Official Review · Reviewer_i49Y · 2022-07-28
**A nice perception dataset and benchmark using radiance fields**

**Rating:** 7
**Confidence:** 4
**Correctness:** Yes.
**Clarity:** Yes.

**Strengths:**

1. The first perception dataset built on radience fields.
2. Significant memory reduction for the storge.
3. Unified representation for 2D and 3D input.

**Weaknesses:**

The title looks quite broad, because the paper only considers indoor scenarios, and benchmark three tasks. The authors are suggested to tone down their claims.

**Additional Feedback:**

See weakness.

**Documentation:**

Yes.

**Ethics:**

No concerns.

**Relation To Prior Work:**

Yes.

**Summary And Contributions:**

This paper builds a novel indoor perception dataset on radience fields, motivated by the recent advances of NeRF. Overall, this is an interesting and novel contribution since it exhibits a notable memory compression rate from the original dataset, and also provides a unified representation of 3D and 2D information. The paper is clearly written and well-organized.

---

> ### Author Response · Authors · 2022-08-22
> **Enhancing the manuscript**
>
> **1. The title looks quite broad because the paper only considers indoor scenarios, and benchmarks three tasks. The authors are suggested to tone down their claims**
>
> Thanks for your valuable suggestion. We will soon update the manuscript to clarify that we benchmark three perception tasks. Note that the PeRFception-CO3D dataset covers both indoor and outdoor environments. We visualized several examples of indoor and outdoor scenes below:
> - PeRFception-CO3D: https://postech-cvlab.github.io/PeRFception/docs/visualization/perf_co3d/
> - PeRFception-ScanNet: https://postech-cvlab.github.io/PeRFception/docs/visualization/perf_scannet/

---

### Author Response · Authors · 2022-08-22
**Shared Response (1/3)**

**[nTsf] The authors provided google drive links for URLs. People in some countries cannot access services provided by Google and hence cannot access this dataset.**
**[FBBw] The authors only provide a toy data download link where data is all from PeRFception-CO3D. I think the authors should provide a part of PeRFception-ScanNet dataset.**
**[FBBW] The documentation is insufficient.**
**[f1Q2] No code is provided, It would be better to provide the source code and full data link.**

As per the reviewer’s requests, we provide our full data, code, and documentation on our project page before the official release. Here are the links:
- Full Data:
  - PeRFception-CO3D Data (Chunks 1): https://1drv.ms/u/s!As9A9EbDsoWcbnHoOoqWmIB6RLs?e=SYGC03
  - PeRFception-CO3D Data (Chunks 2): https://1drv.ms/u/s!AgY2evoYo6FgiwomlG1QUiLg7wqy?e=ReG5Yp
  - PeRFception-ScanNet Data: https://1drv.ms/u/s!AgY2evoYo6FghYVw3MLYwq743fsoUw?e=n56VjA
- Dataset Generation Code: https://github.com/POSTECH-CVLab/PeRFception
- Project Page: https://postech-cvlab.github.io/PeRFception/

We provide OneDrive links and will provide additional ways to access the dataset. Also, we added documentation, additional experiments, command line download scripts, visualizations, and leaderboards on our project page.

**[FBBW] The result on the "object-centric" dataset is not good enough. As the image quality will decrease, I wonder if it is worth this trade-off… Fig.3 needs more discussion. It is desirable that the result trained on PeRFception has a lower accuracy than the original dataset, showing a domain gap caused by reconstruction error**
**[nTsf] This paper creates an impression that the paper is good at data compression. I know this is not the intended purpose of this paper, but the paper reads like this.**

CO3D recently released the second version that fixed a lot of errors in the first version: 1) motion blurs making the reconstruction challenging, 2) repeated images, 3) forward-facing views only scenes, or 4) wrong camera parameters due to textureless background and symmetry of the object. These errors contributed to poor Plenoxel reconstructions.

We created a new PeRFception-CO3D dataset with the CO3D-v2, which contains twice the number of objects and 4X more frames per object with improved image quality. The new PeRFception-CO3D achieves higher fidelity and shows improvement on all metrics on the official toy dataset scenes from CO3D-v2. Especially, we can observe that the LPIPS score, which represents the perceptual similarity, is improved significantly.

|  Method |  PSNR |  SSIM  |  LPIPS |
|:-------:|:-----:|:------:|:------:|
| CO3D v1 | 28.82 | 0.8508 | 0.3534 |
| CO3D v2 | 29.86 | 0.8563 | 0.3179 |

In addition, the data compression rate improved significantly as our 3D reconstruction aggregates redundant information into one canonical space and compresses well while images cannot. Regarding the domain gap, we showed that vision transformer models perform better when trained on our dataset.

| Version | Original Memory (MB) | Ours Memory (MB) | Compression Rate |
|:-------:|:--------------------:|:----------------:|:----------------:|
|    v1   |         75.5         |       69.7       |       -7.7%      |
|    v2   |         144.4        |       71.7       |      -50.3%      |

We visualize several examples in the link
- http://postech-cvlab.github.io/PeRFception/docs/about_paper/v2/

(continued..)

---

> ### Author Response · Authors · 2022-08-22
> **Shareed Response (2/3)**
>
> **[scvN] “The compression step may lose important information for downstream perception tasks. Thus end-to-end perception models should still achieve better accuracy.”**
> **[FBBw] With background augmentation, the accuracy result (test on the original test set) is still lower than trained with the original dataset, as shown in Fig.3.**
> **[scvN] fig 3: there is 4-5% accuracy@1 gap for models trained on co3d and test on co3d / perfception-co3d, which indicates that plenoxels lose info for classification tasks? Also the gap is larger for models trained on PeRFception-CO3D. It makes me doubt the claim "PeRFception-CO3D contains the same information as the original CO3D dataset"**
> **[5TB9] I think there exists information loss after transforming original data to implicit one. A discussion about this is necessary.**
>
> We conduct additional 2D classification experiments using the pre-trained Vision Transformer (ViT) [6] models to show that our PeRFception-CO3D dataset improves the classification performance. Similar to our ResNet 2D classification experiments, we train ViT models on either CO3D images or PeRFception renderings and test on the CO3D test split and the PeRFception-CO3D test split. As reported in the table below, the ViT models trained on the PeRFception-CO3D train set achieve higher classification accuracy than those trained on the CO3D train set. We conjecture that the domain gap for ViT models is much smaller, and the data augmentations on PeRFception made the ViT models to be more robust in real images.
>
> | Train Dataset |  CO3D |       CO3D       | PeRFception-CO3D (bkgd p=0.5) | PeRFception-CO3D (bkgd p=0.5) |
> |:-------------:|:-----:|:----------------:|:-----------------------------:|:-----------------------------:|
> |  Test Dataset |  CO3D | PeRFception-CO3D |              CO3D             |        PeRFception-CO3D       |
> |   ViT/S-16*   | 87.61 |       83.80      |             87.64             |             86.54             |
> |   ViT/L-16*   | 88.30 |       84.90      |             88.65             |             87.59             |
>
> **[FBBW] The choice of Plenoxels needs more discussion. The author says it has explicit geometry and consistent features, but other methods like INGP can also obtain geometry and be trained faster.**
> **[5TB9] The density and color information are also generated by other NeRF-based methods [5, 6, 10, 11]. The 3D structure like points can be generated by Point-NeRF (https://arxiv.org/abs/2201.08845). The reason why we have to use the implicit representation of Plenoxels is not given. A thorough comparison between PeRFception and implicit data generated by methods mentioned before is necessary.**
>
> Among many recent implicit radiance field representations[1,2,3,4,5], we use *Plenoxel* for our data representation for a few reasons—firstly, as we are creating radiance representations for O(10k) scenes, the training and rendering have to be fast for scalability. Ideally, the reconstruction process should take less than an hour, and our Plenoxel-based reconstruction takes 30 min per scene. Secondly, the representation should be able to capture unbounded scenes to represent the backgrounds. Lastly, we want features from the reconstruction to be consistent across scenes for 3D perception tasks. For example, if we train an radiance field MLP per scene, each scene representation or feature would differ from others, and we cannot directly feed the features to a perception network without converting them to other consistent representations. Plenoxels use explicit spherical harmonics features and density, which are consistent across scenes allowing us to train a perception network directly. We compare various radiance fields in the table below for convenience. Note that Plenoxel is the only representation that fits all the criteria.
>
> |   Method  |  Data structure  |  Density |   Color  | Training Time |
> |:---------:|:----------------:|:--------:|:--------:|:-------------:|
> | PointNeRF |    Point Cloud   | Explicit | Implicit |    > 1 day    |
> |    DVGO   |    Dense Grid    | Explicit |  Hybrid  |    < 30min    |
> |  DVGO-v2  |    Dense Grid    | Explicit |  Hybrid  |    < 20min    |
> |    INGP   | Multi-level Hash |  Hybrid  |  Hybrid  |     < 5min    |
> |  TensoRF  |  Decomposed Grid | Explicit |  Hybrid  |    < 30min    |
> | Plenoxels |    Sparse Grid   | Explicit | Explicit |    < 30min    |
>
>
> **[FBBw, scvN] Benchmarking recent state-of-the-art models**
>
>  We have started our revision and are now trying to add more perception models.
> - 2D classification: ViT [6]
> - 3D semantic segmentation: Fast Point Transformer [7]
>
> The results are soon available in the further comments and also in the revised paper.
>
> (continued..)

---

> > ### Author Response · Authors · 2022-08-22
> > **Shared Response (3/3)**
> >
> > **Reference**
> >
> > [1] Sun, Cheng, Min Sun, and Hwann-Tzong Chen. "Direct voxel grid optimization: Super-fast convergence for radiance fields reconstruction." Proceedings of the IEEE/CVF Conference on Computer Vision and Pattern Recognition. 2022.
> >
> > [2] Sun, Cheng, Min Sun, and Hwann-Tzong Chen. "Improved Direct Voxel Grid Optimization for Radiance Fields Reconstruction." arXiv preprint arXiv:2206.05085 (2022).
> >
> > [3] Müller, Thomas, et al. "Instant neural graphics primitives with a multiresolution hash encoding." arXiv preprint arXiv:2201.05989(2022).
> >
> > [4] Chen, Anpei, et al. "TensoRF: Tensorial Radiance Fields." arXiv preprint arXiv:2203.09517 (2022).
> >
> > [5] Xu, Qiangeng, et al. "Point-nerf: Point-based neural radiance fields." Proceedings of the IEEE/CVF Conference on Computer Vision and Pattern Recognition. 2022.
> >
> > [6] Dosovitskiy, Alexey, et al. "An image is worth 16x16 words: Transformers for image recognition at scale." arXiv preprint arXiv:2010.11929 (2020).
> >
> > [7] Park, Chunghyun, et al. "Fast Point Transformer." Proceedings of the IEEE/CVF Conference on Computer Vision and Pattern Recognition. 2022.

---

### Comment · Area_Chair_5Fzr · 2022-08-22
**Discussion**

Dear Reviewers,

Please take a look and respond to author discussions. Let us know if your concerns are alleviated or still present, and if there is a change in your rating.

Thank you, AC

---

> ### Comment · Area_Chair_5Fzr · 2022-08-26
> **Please take a look at author response**
>
> Dear Reviewers, Another reminder to please take a look and respond to author discussions. Let us know if your concerns are alleviated or still present, and if there is a change in your rating.
>
> Best, AC

---

### Author Response · Authors · 2022-08-29
**SOTA 3D semantic segmentation models on PeRFception-ScanNet dataset**

We employ the state-of-the-art large-scale 3D semantic segmentation model based-on transformer architecture, Fast Point Transformer (FPT)[1] , to analyze the performance of the recent transformer-based 3D perception model on our dataset. Identical with the other networks, we use the same hyperparameters for training FPT except the voxel size. We use voxel size of 5cm due to its extensive memory complexity. The results are reported in the below table.
As reported in Table, we observe that FPT can be successfully trained on PeRFception-ScanNet dataset. Moreover, analoguous to the previous observations, we get the best segmentation accuracy when we provide spherical harmonics and density values as input, indicating that those plenoptic features provide additional cues for 3D semantic segmentation task.

| Model | Voxel size (cm) | Feature | mIoU (%) | mAcc (%) |
|-------|-----------------|---------|----------|----------|
| FPT   | 5               | One     |  67.16   |  74.58   |
| FPT   | 5               | D       |  67.99   |  77.35   |
| FPT   | 5               | SH      |  67.42   |  76.71   |
| FPT   | 5               | SH + D  |  68.35   |  77.22   |

[1] Park, Chunghyun, et al. "Fast Point Transformer." Proceedings of the IEEE/CVF Conference on Computer Vision and Pattern Recognition. 2022.

---

### Author Response · Authors · 2022-08-29
**Paper Revision**

Thanks to all the reviewers. We are delighted to enhance our work.
We have revised our main paper and supplementary materials to reflect reviewers' valuable comments. Our main modifications are:

- Discussion on the reason why we’ve selected Plenoxels as our data format.
- Controlled experiments about resolution, quantization methods, and progressive scaling methods.
- Benchmark the recent transformer-based 3D semantic segmentation model, FPT, on PeRFception-ScanNet dataset
- Discussion on the limitations of our work.
- Update scores of PeRFception-ScanNet after enhancing geometries with TSDF initialization.
- Elaborate more details about our experimental setups for clarity.
- Improve readability by adjusting table sizes.
- Fix typo and reflect suggestions about manuscripts.

---

### Meta-Review · Area_Chair_5Fzr · 2022-09-08

**Recommendation:** Accept
**Confidence:** 4

**Metareview:**

Authors propose a novel large-scale implicit representation dataset for perception tasks, called the PeRFception dataset, which consists of two parts that incorporate both object-centric and scene-centric scans for classification and segmentation. The dataset obtains significant memory compression rate while containing both 2D and 3D information in a unified form.

Some important pros from reviews:
+ the first large-scale perception dataset built on radiance fields, implicit representation as another choice for 3D perception
+ significant and impressive storage reduction / compression
+ comprehensive comparison with existing datasets

Authors provided a strong rebuttal, and reviewers agreed that it addressed their concerns. I believe that the dataset is a relevant contribution to the broader research community. Hence, I recommend to accept the paper.
Authors should make sure to incorporate suggestions from reviews into their final manuscript.

---

### Decision · Program_Chairs · 2022-09-16

Accept